# Smoking Status and Risk Awareness of Heated Tobacco Product Use among General Dental Practitioners Belonging to the Aichi Dental Association, Japan

**DOI:** 10.3390/healthcare10122346

**Published:** 2022-11-22

**Authors:** Yukie Oya, Koji Inagaki, Keiji Tokumaru, Toshiyuki Watanabe, Nobuhiro Segawa, Yohei Yamamoto, Shinsuke Takaki, Takahiro Nimi, Makoto Okai, Noriyasu Uchibori, Takahiro Tabuchi, Akio Mitani, Toru Nagao

**Affiliations:** 1Department of Dental Hygiene, Aichi Gakuin University Junior College, Nagoya 464-8650, Japan; 2Department of Periodontology, School of Dentistry, Aichi Gakuin University, Nagoya 464-8651, Japan; 3The Aichi Dental Association, Nagoya 460-0002, Japan; 4Cancer Control Center, Osaka International Cancer Institute, Osaka 541-8567, Japan; 5Department of Maxillofacial Surgery, School of Dentistry, Aichi Gakuin University, Nagoya 464-8651, Japan

**Keywords:** dentists, heated tobacco product, smoking cessation, risk awareness of heated tobacco products, smoking

## Abstract

The awareness of healthcare practitioners concerning heated tobacco product (HTP) use risks has been evaluated; however, few studies have investigated general dental practitioners’ awareness regarding HTP-use risks. In this cross-sectional study, we investigated dentists’ awareness of the risks of smoking, particularly HTP use. A self-administered questionnaire, including eight questions on conventional cigarette and HTP smoking/using status and both knowledge and awareness of HTP-use risks, was posted to 3883 dentists belonging to the Aichi Dental Association, Japan, in August 2019. Statistical analysis was performed using the Statistical Package for Social Sciences; statistical significance was set at *p* < 0.05. We analyzed the data of 1317 dentists (participation rate, 41.6%). The study group included cigarette smokers (11.5%) and HTP users (8.5%), among whom 41.1% were dual users. HTP users were more likely than never smokers/users to correctly perceive HTP-use risks (*p* < 0.05). This study indicates that in Japan, the proportion of HTP users is higher than that of the general population. It is important to educate not only smokers/users but also never smokers/users on the risks of smoking and using HTPs. Smoking cessation, including ceasing HTP use, and aiming to quit smoking and HTP use among dentists would contribute to appropriate smoking cessation among patients.

## 1. Introduction

Smoking prevalence among adults in the United States was approximately 20.0% in 2019, 4.5% of whom were electronic cigarette (e-cigarette) users [1]. Among the current e-cigarette users, the percentage of those who have never smoked cigarettes is the highest among individuals aged 18–24 years (56.0%) [1]. Moreover, as of 2020, 2807 cases of e-cigarette or vaping-use-associated lung injury, including 68 deaths, have been reported, and sale restrictions on any tobacco products, including e-cigarettes, have been in place in the United States since 2019 [2,3].

In contrast, considerable progress has been made in reducing the prevalence of cigarette smoking among adults in Japan over the course of a decade, from 23.4% (38.2% for men and 10.9% for women) in 2009 to 16.7% (27.1% for men and 7.6% for women) in 2019 [4,5]. Nevertheless, heated tobacco products (HTPs), IQOS (IQOS, Philip Morris International, New York, NY, USA), have been marketed since 2014, and glo (glo, British American Tobacco, London, UK) and Ploom TECH (Ploom TECH, Japan Tobacco Inc., Tokyo, Japan) have been distributed since 2016. In 2020, lil Hybrid (lil Hybrid, Philip Morris Japan G.K, Tokyo, Japan) commenced sales in Japan. A total of four HTPs have penetrated the Japanese market. Because of the inaccurate advertisements stating that HTP use does not produce secondhand smoke and is less harmful to health [6], HTP sales have rapidly grown; sales in Japan accounted for 98% of the worldwide share of IQOS in October 2016 [7]. The use of HTPs among current smokers is becoming more popular, particularly among young individuals in their 20s and 30s [4]. Moreover, cases of acute eosinophilic pneumonia following the use of HTPs have been reported in Japan [8,9].

The World Health Organization (WHO) launched a global policy of simplified dental smoking cessation support in 2017, after identifying the occasion of smokers’ visiting a dental healthcare provider as an important opportunity to address smoking cessation before the exacerbation of smoking-related health problems [10]. Smoking cessation interventions should be clinically provided as part of oral health practice. Therefore, dental professionals should provide support for smoking cessation, including HTP-use cessation. Some previous studies have investigated the awareness of healthcare workers, including physicians, medical students, pharmacists, pharmacy students, and dental hygiene students, on the risks of HTP use. However, to the best of our knowledge, no similar studies have investigated such awareness in the field of dentistry, particularly among general dental practitioners [11,12,13,14,15,16].

Thus, this study surveyed the prevalence of smoking and awareness of the risks of smoking, including HTP use, among Japanese dentists, especially those in the Aichi Prefecture, where HTPs were first sold in Japan.

## 2. Materials and Methods

### 2.1. Study Design and Participants

This cross-sectional study included dentists belonging to the Aichi Dental Association, the third-largest dental association in Japan, after the Tokyo and Osaka associations. A self-administered questionnaire was posted to 3883 dentists (3667 male dentists) of the Aichi Dental Association in August 2019.

### 2.2. Sample-Size Calculation

The 2019 National Health and Nutrition Survey showed a smoking rate of 27.1% among men, of which nearly 30% were HTP users [4]. Therefore, this study sample size was calculated with a sample size of 1153, using statistical software EZR [17], assuming a smoking rate of 25% and 95% confidence level.

### 2.3. Study Setting

A questionnaire-based survey was conducted between 20 August and 20 September 2019.

### 2.4. Study Questionnaire

The questionnaire comprised six major items: smoking status, including that related to conventional cigarettes and HTPs; awareness of HTP-use risks; knowledge of the association between periodontal treatment and patients’ smoking status; stages of behavioral change in smoking cessation [18]; importance of smoking cessation in the dental clinic; and the Kano Test for Social Nicotine Dependence. We report two of these items in this paper: smoking status, including that related to conventional cigarettes and HTP use; and awareness of HTP-use risks. The awareness of the harmful effects of HTPs was defined by the respondents’ answers to eight questions (Table 1). Each question had four possible answers (yes/probably yes/no/do not know).

### 2.5. Cigarette-Smoking and/or HTP-Use Status

Cigarette-smoking status was defined as follows:Never smoker: those who had never smoked cigarettes.Former smoker: those who previously smoked cigarettes but have now quit smoking completely.Current smoker: those who smoke cigarettes at least monthly.HTP-use status was defined as follows:Never user: those who had never used HTPs.Former user: those who had previously used HTPs but have now stopped using them completely.Current user: those who use HTPs at least monthly.

Exclusive users of HTPs or conventional cigarettes were defined as participants who only used either one of the two products. Dual users were defined as participants who used both products.

### 2.6. Data Collection

Data were collected through self-administered, anonymous returns by participants. Participants were informed that they could withdraw at any time without giving a reason and that all information would be anonymized and kept strictly confidential. Participants agreed to participate in the study after receiving a written explanation of the study and by completing a questionnaire.

### 2.7. Statistical Analysis

Data were analyzed using Statistical Package for Social Sciences version 28 (IBM Corp, Armonk, NY, USA). To compare the awareness ratio of HTP-use risks and rate of use by demographics, such as sex, age, and smoking status, we used the independent samples chi-square test. Analysis of perceptions of HTP use was performed by using logistic regression analysis after adjusting for sex and age, and never smokers/users were compared to exclusively conventional cigarette, exclusively HTP, and dual user groups. The goodness of fit of the final model was performed by using the Hosmer–Lemeshow test, and odds ratios and 95% confidence intervals were used to calculate the associations. The significance test was a bilateral test, and the level of statistical significance was set at *p*-values < 0.05 for all analyses.

### 2.8. Ethical Considerations

The study was reviewed and approved by the Ethics Committee of Aichi Gakuin University Junior College (No. 19-001). The study considered only completed questionnaires, which indicated consent for participation.

## 3. Results

Data were obtained from 1617 dentists (participation rate, 41.6%); of them, data from 1317 dentists who provided complete information regarding conventional smoking status in the questionnaire were analyzed (94.2% male dentists; response rate, 81.4%) (Table 2). This study sample reached the required sample size. In the statistical analysis, we found differences between the sexes and ages with respect to cigarette-smoking status, HTP-use status, and various other smoking statuses (chi-square test, *p* < 0.05) (Table 3 and Table 4).

### 3.1. Sample Characteristics

The participants were mostly older than 50 years (20–49 years: 30.5%, ≥50 years: 69.3%, and not answered: 0.2%). Most of the participants were practicing dentists (94.0%).

### 3.2. Conventional Cigarette-Smoking Status

The participants included 42.9% never smokers (n = 565), 45.6% former smokers (n = 600), and 11.5% current smokers (n = 152). Smoking status was the highest among men and those in the 50–59-year-old age group (*p* < 0.05) (Table 3). Current cigarette smokers smoked an average of 14.0 (±7.7) (range: 1–40) cigarettes per day and had an average smoking history of 18.4 years (±12.5 years) (range: 1–50 years).

### 3.3. HTP-Use Status

The rate of never, former, and current HTP users was 64.8% (n = 854), 4.4% (n = 58), and 8.5% (n = 112), respectively, but 22.2% (n = 293) of the dentists did not answer this question (Table 4). HTP-use status was the highest among men and those in the 50–59-year-old age group (*p* < 0.05). Among the 112 current HTP users, 41.1% (n = 46) were dual users who also currently smoked cigarettes, whereas 58.9% (n = 66) used only HTPs (Table 5). More than half of the current HTP users had previously smoked conventional cigarettes (58.0%). Among the HTP users, 4.5% (n = 5) used three products (IQOS, glo, and PloomTECH), and 10.7% (n = 12) used two products (IQOS and glo, IQOS and PloomTECH, or glo and PloomTECH).

### 3.4. Risk Awareness of HTP Use

The results of the analyses examining the participants’ awareness of HTP-use risk are presented in Table 6. On the basis of the odor and safety of HTP among users, current smokers/users were more likely to accurately perceive the risks of HTP use than were never smokers/users (*p* < 0.05). Moreover, exclusively HTP users were more likely than never smokers/users to consider HTPs as substitutes for conventional cigarettes (*p* < 0.05).

## 4. Discussion

### 4.1. Conventional Cigarette-Smoking Status

Although large-scale studies have been conducted among physicians, no comparable large-scale study has been reported among dentists. The cigarette-smoking rate among dentists in this study was 11.5% in 2019, which was lower than that reported in the 2019 National Health and Nutrition Examination Survey (16.7%) and in the Aichi Prefecture in 2016 (15.6%) [4,19]. However, this value was higher than that among members of the Japanese Society of Periodontology in 2013 (8.4%), members of the Japanese Society of Oral and Maxillofacial Surgeons in 2018 (7.2%), and physicians in Japan in 2020 (7.1% for men and 2.1% for women) [20,21,22]. Previous studies in Japan have also reported that the conventional cigarette-smoking statuses among physicians, dentists, and dental hygienists were 7.8%, 1.3%, and 0.6%, respectively [21,23]. In addition, the smoking rate in this study was far higher than that among physicians, dentists, and dental hygienists in other countries [11,23]. In addition to the fact that more than 90% of the dentists surveyed were men, this group had a significantly higher percentage of male smokers, which may have contributed to the higher smoking rate compared with that of other medical professionals. To obtain basic data on the smoking pattern and behavior among dentists, a larger scale survey of dentists is warranted.

### 4.2. Status and Pattern of HTP Use

This study is among the few that have investigated the use of HTP among dentists. The present study researched the reality of complex forms of smoking, including HTP, in Japan, where awareness of HTPs has generally increased in recent years [24,25]. The rate of current HTP use among dentists was 8.5%, with the highest HTP use among men aged 50−59 years, consistent with conventional cigarettes and the distribution of HTP users. Moreover, 41.1% of HTP users used both HTPs and conventional cigarettes. For this reason, we considered that the results were influenced not only by the tendency of those with conventional cigarette-smoking experience to initiate HTP use but also by the fact that both conventional cigarettes and HTPs were significantly smoked by men aged 50−59 years, a group who smoked significantly more than the other groups. In Canada, England, the United States of America, and Australia, the use of HTPs among individuals aged ≥18 years has been reported to range between 0.2% and 1.2% [26]. Surprisingly, our study demonstrated that HTP use in 2018 was more prevalent among dentists in Japan (8.5%) than among the general adult (≥20 years of age) population in Japan (2.7%) [27]. Although brief tobacco interventions by oral health professionals have been recommended by the WHO since 2017 [10], the HTP-use status of dentists in Japan was higher than that of the general population. Physicians’ smoking status contributes to reduced motivation for initiating cessation interventions and affects cessation interventions in patients who smoke [28,29]. Thus, the dentists’ HTP-use status in this study could be associated with a reduced likelihood of smoking cessation among patients who receive dental treatment at dental clinics. Future studies should be conducted to determine the relationship between HTP-use status in dentists and the current status of smoking cessation in dental clinics.

Additionally, this study showed that most current HTP users were former conventional cigarette smokers, whereas only 0.9% of HTP users had never smoked conventional cigarettes. This result is consistent with that of a study that reported that a greater proportion of HTP users had smoked or had experience with conventional cigarettes [27]. HTP use could interfere with smoking cessation. Furthermore, it is necessary to highlight the negative effects of smoking on oral health and encourage the understanding of tobacco issues, including HTP use, among healthcare professionals involved in providing smoking cessation support.

### 4.3. Participants’ Awareness of Risks Associated with HTP Use

Although the content of nicotine and other carcinogens in mainstream HTP smoke is lesser than that in cigarettes, this does not guarantee a reduction in the incidence of health problems of users or those who are exposed to secondhand HTP smoke [6]. Nevertheless, HTPs are mistakenly perceived as products with reduced adverse health effects [29,30,31,32,33]. Previous research on correct perceptions of HTPs in a survey of medical students in Poland revealed that individuals who had never used HTPs were significantly more likely than HTP users to perceive health problems, smoking location, and secondhand smoking issues associated with HTPs [12]. Moreover, individuals who had never used HTPs were significantly more aware of the issues related to health problems, smoking cessation, and a reduction in cigarette consumption, as reported in a survey among 14–25-year-old individuals in Hong Kong [34]. Contrarily, in this study, conventional cigarette smokers and HTP users in Japan were significantly more likely than never users to correctly perceive issues related to odor and health problems. For the item “substitute for conventional cigarettes”, exclusively HTP users were more likely than never users to have misperceptions. Few studies have investigated the correct perception of HTPs; moreover, no previous studies have demonstrated that HTP users have more-accurate perceptions of the risks of HTPs than never users, as in the present study. For HTP users, we speculate that the nicotine contained in tobacco and conventional cigarettes affected the individuals’ physical dependence and that they continued to use the products despite being aware of the risks of HTP use. In addition, most exclusively HTP users in this study were former smokers who seemed to be using the product as a substitute for conventional cigarettes. However, the frequency of smoking increases when switching to HTPs compared with when smoking conventional cigarettes [32]. Similarly, in the present study among dentists in Japan, HTP users themselves used HTPs under the perception that they are a substitute for conventional cigarettes; however, there is a concern that the frequency of smoking would increase compared with that with conventional cigarette smoking. Hence, it is possible that the health problems possibly arising from the use of HTPs may become more severe because of multiple inhalations of nicotine and other harmful substances. Additionally, Philip Morris International, of the tobacco industry, has used dental health professionals to advertise tobacco products, promoting the switch from conventional cigarettes to IQOS [35]. We should note that the tobacco industry is currently targeting Japanese dental professionals to spread misperceptions through a series of inaccurate advertising activities. Therefore, in future surveys, the frequency of HTP use should be added to the survey items and compared to that of conventional cigarette smoking. Furthermore, because the level of awareness among never users was low in this study, we think that it is necessary to impart precise knowledge regarding HTPs to never users to promote smoking cessation in dentists as medical professionals involved in smoking cessation support.

### 4.4. Study Limitations

This study had some limitations. First, approximately one-third of the participants who received the questionnaire responded by mail. If the data had been obtained from those who did not respond, the results of the present analysis would have changed. Second, although the questionnaire in the present study was completed anonymously, selection bias or other factors could have affected the generalization of the results. Third, the respondents in this study were only dentists with clinical experience, most of whom were men, which was due to the absence of potential confounders, such as socioeconomic status, education level, etc. This may have influenced the results. Fourth, the peculiarity of the area where this research was conducted, Aichi Prefecture in Japan, could be a possible limitation. The IQOS, an HTP, was sold in Japan before it was sold in the rest of the world, and Aichi Prefecture was the first region in Japan where IQOS was sold. Furthermore, sales of IQOS in Japan accounted for more than 98% of the global market share in 2016 [7]. This may have influenced the risk awareness of HTPs among smokers in this study.

To compensate for these limitations, a similar survey should be conducted on dental students and qualified dentists in other prefectures and countries for further comparison in the future.

## 5. Conclusions

Oral healthcare professionals are among the few healthcare professionals who are in contact not only with patients with health problems caused by smoking but also with patients of all ages who do not have smoking-related health problems at the time of their visit. For the dental clinic/office to be a venue for smoking cessation support, first educating dentists on smoking cessation, including the cessation of HTP use, is essential. Furthermore, it is necessary to educate not only smokers/users but also never smokers/users regarding the risks of HTP use and conventional cigarette smoking to support the health of patients.

## Figures and Tables

**Table 1 healthcare-10-02346-t001:** Questionnaire on risk awareness of heated tobacco product use.

Question Item
Do you think that HTPs are odorless?
Do you think that HTPs are safe for the air?
Do you think that HTPs are harmless?
Do you think that HTPs are safe for secondhand smokers’ health?
Do you think that HTP use is smoking?
Do you think that HTPs can be used in a smoke-free area?
Do you think that HTP use can help with smoking cessation?
Do you think that HTPs are a substitute for conventional cigarettes?

HTP: heated tobacco product.

**Table 2 healthcare-10-02346-t002:** Participants’ characteristics.

		Total		Men		Women	
		(n = 1317)		(n = 1240)		(n = 77)	
		n	%	n	%	n	%
Age (years)	20–29	6	0.5	6	0.5	-	-
	30–39	119	9.1	111	9	8	10.5
	40–49	276	21	253	20.4	23	30.3
	50–59	395	30.1	373	30.1	22	28.9
	60–69	406	30.9	388	31.3	18	23.7
	≥70	112	8.5	107	8.6	5	6.6
Dental education level							
Nonspecialist		1040	80.7	974	80.4	66	85.7
Specialist		248	19.3	237	19.6	11	14.3
State of employment							
General practitioner		1238	94.2	1169	94.5	69	94.5
Working as employees		70	5.3	70	5.1	7	5.1
Leave of absence		6	0.5	6	0.4	1	0.4

Nonspecialist: Did not specialize in a particular dental field; no response: age (n = 3), dental education level (n = 29); state of employment (n = 3).

**Table 3 healthcare-10-02346-t003:** Prevalence of conventional cigarette smoking.

Variable	Category	Current Smoker		Former Smoker	Never Smoker	*p*
		(n = 152)				
		Dual User	Exclusively Cigarette Smoker			
		(n = 46)	(n = 106)	(n = 600)	(n = 565)	
		% (95% Confidence interval)				
Prevalence		3.5 (2.6–4.6)	8.0 (6.6–9.5)	45.6 (42.9–48.3)	42.9 (40.3–45.6)	-
Demographics						
Sex	Male	100 (96.0–100.7)	98.1 (93.0–99.9)	98.2 (96.7–99.0)	88.7 (85.8–91.0)	0.001
	Female	-	1.9 (0.1–7.0)	1.8 (1.0–3.3)	11.3 (9.0–14.2)	
Age (years)	20–29	-	-	0.3 (0.0–1.3)	0.7 (0.2–1.9)	0.001
	30–39	13.0 (5.7–26.0)	6.6 (2.5–11.2)	6.2 (4.5–8.4)	12.2 (9.8–15.2)	
	40–49	32.6 (20.8–47.1)	10.4 (4.9–16.0)	17.8 (14.9–21.0)	25.5 (22.1–29.2)	
	50–59	34.8 (22.6–49.3)	42.5 (33.0–52.3)	30.2 (26.6–34.0)	27.3 (23.7–31.1)	
	60–69	17.4 (8.8–31.0)	26.4 (18.5–35.0)	35.7 (31.9–39.6)	27.8 (24.3–31.6)	
	≥70	2.2 (−0.7 to 12.4)	14.2 (7.5–21.4)	9.9 (7.7–12.6)	6.5 (4.8–8.9)	

Chi-square tests were used to account for the complex survey design. SD: standard deviation.

**Table 4 healthcare-10-02346-t004:** Prevalence of heated tobacco product use.

Variable	Category	Current User		Former User	Never User	*p*
		(n = 112)				
		Dual User	Exclusively HTP User			
		(n = 46)	(n = 66)	(n = 58)	(n = 854)	
		% (95% Confidence interval)				
Prevalence		3.5 (2.6–4.6)	5.0 (4.0–6.3)	4.4 (3.3–5.6)	64.8 (62.1–67.7)	-
Demographics						
Sex	Male	100 (96.0–100.7)	100 (93.4–101.0)	98.3 (93.8–100.0)	91.8 (90.0–93.5)	0.018
	Female	-	-	1.7 (0.0–6.3)	8.2 (6.5–10.0)	
Age (years)	20–29	-	-	-	0.6 (0.1–1.2)	0.001
	30–39	13.0 (5.7–26.0)	16.7 (9.4–27.6)	13.8 (5.0–23.5)	10.4 (8.5–12.7)	
	40–49	32.6 (20.8–47.1)	39.4 (28.5–51.5)	24.1 (13.5–35.3)	23.1 (20.4–26.0)	
	50–59	34.8 (22.6–49.3)	37.9 (27.1–50.0)	36.2 (24.6–49.2)	29.4 (26.5–32.5)	
	60–69	17.4 (8.8–31.0)	6.1 (1.9–15.0)	20.7 (10.0–31.6)	29.3 (26.4–32.2)	
	≥70	2.2 (−0.7 to 12.4)	-	5.2 (0.0–11.1)	7.2 (5.4–9.0)	
Device brand preferences						
Device brand	IQOS	47.8 (34.1–61.9)	69.7 (57.7–79.5)	-	-	-
	glo	21.7 (12.1–35.8)	22.7 (14.2–34.3)			
	Ploom TECH	47.8 (34.1–61.9)	28.8 (19.2–40.7)			
Number of products	1	87.0 (74.0–94.3)	83.4 (72.4–90.6)	-	-	-
	2	8.7 (2.9–20.9)	12.1 (6.0–22.4)			
	3	4.3 (0.4–15.3)	4.5 (1.0–13.0)			

Chi-square tests were used to account for the complex survey design. HTP: heated tobacco product; no response (n = 293).

**Table 5 healthcare-10-02346-t005:** Prevalence of heated tobacco product use and user characteristics.

Variable	Category	HTP Use by Conventional Cigarette-Smoking Status			*p*
		Dual User	Exclusively HTP User		
		Current Smoker	Former Smoker	Never Smoker	
		(n = 46)	(n = 65)	(n = 1)	
		% (95% Confidence interval)			
Prevalence		41.1 (32.4–50.3)	58.0 (48.8–66.8)	0.9 (−0.3–5.4)	-
Demographics					
Sex	Male	100 (90.8–101.5)	100 (93.3–101.1)	100 (16.8–103.9)	-
	Female	-	-	-	
Age (years)	20–29	-	-	-	0.540
	30–39	13.0 (5.7–26.0)	16.9 (9.5–28.0)	-	
	40–49	32.6 (20.8–47.1)	38.5 (27.6–50.6)	100 (16.8–103.9)	
	50–59	34.8 (22.6–49.3)	38.5 (27.6–50.6)	-	
	60–69	17.4 (8.8–31.0)	6.2 (2.0–15.2)	-	
	≥70	2.2 (−0.7 to 12.4)	-	-	
Device brand preferences					
Device brand	IQOS	47.8 (34.1–61.9)	70.7 (58.7–80.5)	-	-
	glo	21.7 (12.1–35.8)	21.5 (13.2–33.1)	100 (16.8–103.9)	
	PloomTECH	47.8 (34.1–61.9)	29.2 (19.5–41.3)	-	
Number of products	1	87.0 (74.0–94.3)	83.1 (72.0–90.5)	100 (16.8–103.9)	-
	2	8.7 (2.9–20.9)	12.3 (6.1–22.7)	-	
	3	4.3 (0.4–15.3)	4.6 (1.1–13.2)	-	

Chi-square tests were used to account for the complex survey design. HTP: heated tobacco product.

**Table 6 healthcare-10-02346-t006:** Multiple logistic regression examining correct risk awareness of heated tobacco product use by smoking/using status.

Statement	Dual User		Exclusively Cigarette Smoker		Exclusively HTP User	
	(n = 46)		(n = 106)		(n = 66)	
	(Ref: Never Smoker/User)					
	YES (%)/NO (%)	aOR (95% CI)	YES (%)/NO (%)	aOR (95% CI)	YES (%)/NO (%)	aOR (95% CI)
HTPs are odorless (REF: Yes)	46/54	1.9 * (1.0–3.6)	57/43	1.7 * (1.1–2.6)	49/51	1.5 (0.9–2.5)
HTPs are safe for the air (REF: Yes)	48/52	0.9 (0.5–1.8)	45/55	1.5 (1.0–2.4)	38/62	1.2 (0.7–2.1)
HTPs are harmless (REF: Yes)	20/80	2.4 * (1.1–5.2)	33/67	1.7 * (1.1–2.7)	15/85	2.7 ** (1.3–5.6)
HTPs are safe for secondhand smokers’ health (REF: Yes)	26/74	1.9 (0.9–3.7)	42/59	1.3 (0.8–2.1)	27/73	1.4 (0.8–2.6)
HTP use is smoking (REF: No)	52/48	1.6 (0.9–3.0)	36/64	1.1 (0.7–1.7)	47/53	1.2 (0.7–2.0)
HTPs can be used in a smoke-free area (REF: Yes)	35/65	1.2 (0.6–2.3)	39/61	1.4 (0.9–2.2)	27/73	1.6 (0.9–2.9)
HTP use can help with smoking cessation (REF: Yes)	33/67	1.8 (0.9–3.5)	50/50	1.2 (0.8–1.9)	36/64	1.4 (0.8–2.4)
HTPs are a substitute for conventional cigarettes (REF: Yes)	63/37	1.0 (0.5–1.8)	63/37	1.1 (0.7–1.8)	80/20	0.4 ** (0.2–0.8)

* *p* < 0.05, ** *p* < 0.01; Adjusted odds ratios (aORs) represent the results of the multivariable logistic regression analysis adjusted for sex and age. CI: confidence interval, HTP: heated tobacco product, REF: reference value.

## Data Availability

The data presented in this study are available on request from the corresponding authors.

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
