# Peer review of "Smoking Status and Risk Awareness of Heated Tobacco Product Use among General Dental Practitioners Belonging to the Aichi Dental Association, Japan"

_healthcare, 2022, doi:10.3390/healthcare10122346_

Round 1

Reviewer 1 Report

In my opinion the paper can be informative and provide a valuable source document for anyone requiring a primer to know and understand this issue. But, numerous shortcomings in the section Methods, Results and Discussion make this paper not appropriate for publication in this form and significant corrections should be made (major revision). Some comments:  

  • Lines 75: Correct and reconstruct the section so that you list the following subsections: Study design, Study setting, Study population, Study Sample, Sample size calculation, Data collection, Questionnaire, Statistical analysis, Ethical consideration. State from when until when did data collection last. 
  • Line 107-112: Describe in detail the used statistical methods, in particular the logistic regression analysis, adjusting, indicators presented in this manuscript. 
  • Lines 117-120: Inscribe the `Participation rate` and `Response rate` in this study. State the reasons for not participating in the study, State whether the analysis included questionnaries that were not completely filled out. State how the issue of missing data was handled. 
  • Table 2: Line 119 states that `1,317 dentists who provided complete information regarding conventional smoking status in the questionnaire were analyzed`, while on the Table 2 the number for Total=1423. Explain. 
  • Line 129: The number of participants on Table 2 (Total=1423) and on Table 3 (Total=1090) differ significantly.  
  • Lines 129-131: Must add one new table that would present `Sample Characteristics`, with age, sec, education level distribution, etc. Following that description, present Tables with the results about tobacco use. 
  • Lines 132-137: It is not good practice to just rewrite data from Table 2, and especially not with providing a too broad description of these data. Correct the text of that paragraph so that it emphasizes where statistically significant differences were found. Alike Table 3, state the number (%) of dual smokers. 
  • Lines 138-147: It is not good practice to just rewrite data from Table 3, and especially not with providing a too broad description of these data. Correct the text of that paragraph so that it emphasizes where statistically significant differences were found.
  • Line 177: Explain the notable differences in cigarette smoking prevalence in this study in comparison to others. 
  • Line 180-181: Explain `trends` and where in this manuscript it was described. 
  • Lines 181-183: Pay attention that the section Results presented data about the prevalence of tobacco use, and not incidence. Explain and correct. 
  • Lines 196-198: It is not good practice when writing in the section Discussion to cite a Table from the section Results. 
  • Line 203: Special attention in the section Discussion should be paid to age and sex distribution, both in conventional and HTP use, where statistically significant differences were presented in this manuscript. Give a comparison with results of other studies and an explanation for the found differences.  
  • Lines 205-207: Pay attention to the fact that the section Results presented data about the prevalence of tobacco use, and not incidence. Explain and correct. 
  • Lines 205-207: Cite the appropriate reference. 
  • Lines 209-212: Cite the appropriate reference. 
  • Line 256: A particular problem is the small number of variables (only age and sex) that are described in this manuscript. A limitation is the absence of potential confounders in the analysis in this study (socioeconomic status, state or private dentist, education level, etc).  
  • Lines 260-272: Reconstruct the section Conclusions so that the results obtained in this manuscript are emphasized, that is - for tobacco use in dentists. This manuscript does not present data for tobacco use in the general population, so the question is whether mentioning them is justified in the conclusion. 

Author Response

November 10, 2022

Dr. Rahman Shiri

Editor-in-Chief

Healthcare

Dear Editor:

On behalf of my co-authors, I am submitting the revised manuscript titled “Smoking status and risk awareness of heated tobacco product use among dentists belonging to the Aichi Dental Association in Japan.”

We greatly appreciate your support as well as the insightful comments of the two reviewers. The manuscript has been revised according to the suggestions of the reviewers as detailed below. The revised parts are marked up in our revised manuscript using the “Track Changes” function.

We look forward to working with Healthcare to ensure that the published manuscript meets your rigorous intellectual and technical standards.

Thank you for your consideration. I look forward to hearing from you.

Sincerely,

Koji Inagaki

Department of Dental Hygiene Aichi Gakuin University, Junior College

1-100 Kusumoto-cho, Chikusa-ku, Nagoya, 464-8650, Japan

Phone number: +81-52-751-2570

Fax number: +81-52-761-3461

Email address: [email protected]

Reviewer #1: In my opinion the paper can be informative and provide a valuable source document for anyone requiring a primer to know and understand this issue. But, numerous shortcomings in the section Methods, Results and Discussion make this paper not appropriate for publication in this form and significant corrections should be made (major revision). Some comments:

  1. Lines 75: Correct and reconstruct the section so that you list the following subsections: Study design, Study setting, Study population, Study Sample, Sample size calculation, Data collection, Questionnaire, Statistical analysis, Ethical consideration. State from when until when did data collection last.

AUTHORS: Thank you for your valuable suggestion. We listed the following descriptions: Study design, Study setting, Study population, Data collection (p. 2, lines 76–80), Questionnaire (p. 2, lines 81–92), Statistical analysis (p. 3, lines 107-113), and Ethical consideration (p. 3, lines 114-117). Considering sample size calculation, we did not calculate the sample size in this cohort study and decided to include all the dentists affiliated with the Aichi Dental Association.

  1. Line 107-112: Describe in detail the used statistical methods, in particular the logistic regression analysis, adjusting, indicators presented in this manuscript.

AUTHORS: Thank you for your valuable suggestion. We have supplemented the statistical analysis section with explanations (p. 3, lines 107–113).

  1. Lines 117-120: Inscribe the `Participation rate` and `Response rate` in this study. State the reasons for not participating in the study, State whether the analysis included questionnaries that were not completely filled out. State how the issue of missing data was handled.

AUTHORS: We have revised the text to reflect your concerns (p. 3, lines 119–122). We analyzed only the data providing complete information regarding conventional smoking status in the questionnaire (p. 3, lines 119–122).

  1. Table 2: Line 119 states that `1,317 dentists who provided complete information regarding conventional smoking status in the questionnaire were analyzed`, while on the Table 2 the number for Total=1423. Explain.
  2. Line 129: The number of participants on Table 2 (Total=1423) and on Table 3 (Total=1090) differ significantly.
  3. Lines 132-137: It is not good practice to just rewrite data from Table 2, and especially not with providing a too broad description of these data. Correct the text of that paragraph so that it emphasizes where statistically significant differences were found. Alike Table 3, state the number (%) of dual smokers.
  4. Lines 138-147: It is not good practice to just rewrite data from Table 3, and especially not with providing a too broad description of these data. Correct the text of that paragraph so that it emphasizes where statistically significant differences were found.

AUTHORS: Considering Tables 2 and 3, since exclusive smokers/users exist among current smokers/users, the figures have not been revised. However, since the sentence was misleading, the tables have been revised to make it easier to understand, listing details regarding the current smokers/users, exclusive smokers/users, and dual users (p. 4, 5, lines 127-131, Tables 3 and 4).

  1. Lines 129-131: Must add one new table that would present `Sample Characteristics`, with age, sec, education level distribution, etc. Following that description, present Tables with the results about tobacco use.

AUTHORS: Thank you for your valuable suggestion. We have added a new Table 2 (p. 4, lines 125, 126), which outlines the participants’ characteristics such as sex, age, smoking status, dental education level, and state of employment.

  1. Line 177: Explain the notable differences in cigarette smoking prevalence in this study in comparison to others.

AUTHORS: Thank you for your valuable suggestion. We have added the word "far" for emphasis to clarify the sentence (p. 7, lines 175–178).

  1. Line 180-181: Explain `trends` and where in this manuscript it was described.

AUTHORS: We have added the appropriate references (p. 7, lines 180–182).

  1. Lines 181-183: Pay attention that the section Results presented data about the prevalence of tobacco use, and not incidence. Explain and correct.

AUTHORS: Thank you for your valuable suggestion. We have changed the word from “incidence” to “rate” (p. 7, lines 182–183).

  1. Lines 196-198: It is not good practice when writing in the section Discussion to cite a Table from the section Results.

AUTHORS: Thank you for pointing this out. We completely agree with your point and have omitted “Table 2” citation.

  1. Line 203: Special attention in the section Discussion should be paid to age and sex distribution, both in conventional and HTP use, where statistically significant differences were presented in this manuscript. Give a comparison with results of other studies and an explanation for the found differences.

AUTHORS: We have incorporated this comment and compared it with other studies based on the age and sex distribution, which showed significant differences in this study (p. 7, lines 184–186).

  1. Lines 205-207: Pay attention to the fact that the section Results presented data about the prevalence of tobacco use, and not incidence. Explain and correct.

AUTHORS: Thank you for your valuable suggestion. We have revised the word “prevalence” to “rate” (p. 5, lines 142–144).

  1. Lines 205-207: Cite the appropriate reference.

AUTHORS: We have added the appropriate reference (p. 8, lines 208–210).

  1. Lines 209-212: Cite the appropriate reference.

AUTHORS: We have added the appropriate reference (p. 8, lines 210–215).

  1. Line 256: A particular problem is the small number of variables (only age and sex) that are described in this manuscript. A limitation is the absence of potential confounders in the analysis in this study (socioeconomic status, state or private dentist, education level, etc).

AUTHORS: Thank you for your valuable suggestion. We agree with your suggestion; thus, some of the following potential confounders have been listed in Table 2: Dental education level and State of employment (p. 4, lines 125–126, Table 2). Moreover, we have added a new limitation, which was the absence of potential confounders such as socioeconomic status, education level, etc.

  1. Lines 260-272: Reconstruct the section Conclusions so that the results obtained in this manuscript are emphasized, that is - for tobacco use in dentists. This manuscript does not present data for tobacco use in the general population, so the question is whether mentioning them is justified in the conclusion.

AUTHORS: Thank you for your valuable suggestion. We have clarified that “general population” indicates “general Japanese population” (the national health and nutrition survey in Japan) and added the word "extremely" for emphasis to clarify the sentence (p. 9, lines 263–265).

Reviewer 2 Report

Thank you for the opportunity to review your manuscript. I ask the authors to correct the following:

- please state the hypothesis or research question.

- What were the eligibility criteria for participants? Please describe this in detail. Why was this group considered representative?

- The descriptions of the tables (or rather the lack of them) are unacceptable. Please describe the results in detail with percentages, numbers, test scores, correlation coefficients and probability levels. Descriptions can be a repetition of what is in the tables, especially since tables may not be clear to everyone.

- Please clarify under the tables what the values in parentheses represent.

- P please write as p-value in the tables.

Author Response

November 10, 2022

Dr. Rahman Shiri

Editor-in-Chief

Healthcare

Dear Editor:

On behalf of my co-authors, I am submitting the revised manuscript titled “Smoking status and risk awareness of heated tobacco product use among dentists belonging to the Aichi Dental Association in Japan.”

We greatly appreciate your support as well as the insightful comments of the two reviewers. The manuscript has been revised according to the suggestions of the reviewers as detailed below. The revised parts are marked up in our revised manuscript using the “Track Changes” function.

We look forward to working with Healthcare to ensure that the published manuscript meets your rigorous intellectual and technical standards.

Thank you for your consideration. I look forward to hearing from you.

Sincerely,

Koji Inagaki

Department of Dental Hygiene Aichi Gakuin University, Junior College

1-100 Kusumoto-cho, Chikusa-ku, Nagoya, 464-8650, Japan

Phone number: +81-52-751-2570

Fax number: +81-52-761-3461

Email address: [email protected]

Reviewer #2: Thank you for the opportunity to review your manuscript. I ask the authors to correct the following:

  1. please state the hypothesis or research question.

AUTHORS: Thank you for your valuable suggestion. We have clarified the hypothesis or research question (p. 2, lines 66–73).

  1. What were the eligibility criteria for participants? Please describe this in detail. Why was this group considered representative?

AUTHORS: We have defined the eligibility criteria for participants being dentists belonging to the Aichi Dental Association (p.2, lines 76–80). We considered the Aichi Prefecture to be a representative group because it was the first place in Japan where HTPs were sold. We have also mentioned this in the manuscript (p. 2, lines 71–73).

  1. The descriptions of the tables (or rather the lack of them) are unacceptable. Please describe the results in detail with percentages, numbers, test scores, correlation coefficients and probability levels. Descriptions can be a repetition of what is in the tables, especially since tables may not be clear to everyone.

AUTHORS: Thank you for your valuable suggestion. We have explained the results in detail with percentages, numbers, test scores, correlation coefficients, and probability levels.

  1. Please clarify under the tables what the values in parentheses represent.

AUTHORS: Thank you for your valuable suggestion. We have clarified that the values in the parentheses in the table are 95% confidence intervals.

P please write as p-value in the tables.

AUTHORS: Thank you for your valuable suggestion. We have changed the word from “P” to “p-value” in all the tables.

Round 2

Reviewer 1 Report

The lines authors state throughout their response letter do not match the text in the provided pdf document. This made the assessment of implemented changes more difficult. However, once the revised version is compared to the previous version - it is clear that at numerous occasions authors stated that they made changes which in fact they did not, which is to say the least very misleading, therefore those issues still remain to be resolved. Comments are as follows:

- The authors did not correct and reconstruct the Methods section, even though they stated in their reply that they listed the suggested subsections. Once again, the current structure of the section Methods is not in the form of usual comprehensive reporting of the methods, it does not contain usual subsections necessary to enhance readability and reproducibility. It is not clear why the authors indicated they made a change they did not.

- No details regarding the logistic regression analysis were added, yet the authors stated that they added explanations regarding this. - The question about the sample size remains. The authors replied that they did not calculate the sample size as they decided to include all dentists affiliated with the mentioned association. However, the stated response and participation rates were very low, around and less than 40%. This brings up the issue of whether this study was powered enough to detect meaningful and significant differences. If you state that "Considering sample size calculation, we did not calculate the sample size in this cohort study and decided to include all the dentists affiliated with the Aichi Dental Association." and you have a participation rate of 41.6% then that is not a satisfactory sample size for this type of study. - Regarding the question of missing data, and in line with your respone - you defined a questionnaire as complete if it contained complete information on conventional smoking status, but what about all of the other questions? Explain. - Results: Almost no changes were made to the text, and none in line with the received comments regarding the description of results given on tables, nor was a rationale provided. - My comment on the original version of the manuscript was: "Line 177: Explain the notable differences in cigarette smoking prevalence in this study in comparison to others.". The Authors' reply is that they added the word "far" to another sentence. This does not represent a response to this comment. No explanation was provided. Simply listing results of this and other studies does not represent a discussion nor provides any meaningful information. Providing possible explanations for any noted differences is the main part of the section discussion. - My comment on the original version of the manuscript was: "Line 180-181: Explain `trends` and where in this manuscript it was described.". The authors responded by adding two references. This is misleading - the sentence states that the present study analyzed trends - explain where. - My comment on the original version of the manuscript was: "Line 203: Special attention in the section Discussion should be paid to age and sex distribution, both in conventional and HTP use, where statistically significant differences were presented in this manuscript. Give a comparison with results of other studies and an explanation for the found differences." was not answered appropriately, even though authors stated they did. No explanations were provided, and comparisons are made with just one study even though "studies" are mentioned by the authors. - In a response to my comment, authors stated that they: "Moreover, we have added a new limitation, which was the absence of potential confounders such as socioeconomic status, education level, etc.". However, no changes were made to the section Limitations - this is misleading. - The Authors' response to the last comment regarding the Conclusion is extremely misleading - the authors completely ignored the provided remarks, interpreted them wrongly in a way that enabled them to make no changes other than reorder the words and add a word to make it seem as if they have allegedly answered the comment. The issue still remains - instead of underlining their own results, the authors make unjustified comparisons and conclusions that are no where related to their results. 

Author Response

Reviewer #1: The lines authors state throughout their response letter do not match the text in the provided pdf document. This made the assessment of implemented changes more difficult. However, once the revised version is compared to the previous version - it is clear that at numerous occasions authors stated that they made changes which in fact they did not, which is to say the least very misleading, therefore those issues still remain to be resolved. Comments are as follows:

Response: We apologize for any inconvenience caused. We have matched the PDF document with the number of lines in the PDF document, since the number of lines in a PDF document and a document using the “Track Changes” function is different.

  1. The authors did not correct and reconstruct the Methods section, even though they stated in their reply that they listed the suggested subsections. Once again, the current structure of the section Methods is not in the form of usual comprehensive reporting of the methods, it does not contain usual subsections necessary to enhance readability and reproducibility. It is not clear why the authors indicated they made a change they did not.

Response: We regret that we were unable to adequately correct the problem despite your suggestion. We have reviewed it again based on your suggestion and have revised the subsections as follows: Study design and Participants (p. 2, lines 75–79), Sample Size Calculation (p. 2, lines 80–84), Study setting (p. 2, lines 85–87), Study Questionnaire (p. 2,3 lines 88–99), Cigarette Smoking and/or HTP Use Status (p. 3, lines 100–113), Data collection (p. 3, lines 114–119), Statistical Analysis (p. 3, lines 120−130), and Ethical Consideration (p. 3,4, lines 131−134).

  1. No details regarding the logistic regression analysis were added, yet the authors stated that they added explanations regarding this.

Response: We apologize for the inadequate explanation. Accordingly, we have added the following text to the Statistical Analysis (p. 3, lines 124−128): Analysis of perceptions of HTP use was performed using logistic regression analysis after adjusting for sex and age, and never smoker/user were compared to exclusive conventional cigarette, exclusive HTP, and dual user groups. The goodness of fit for the final model was performed using the Hosmer–Lemeshow test, and the odds ratios and 95% confidence intervals were used to calculate the associations.

  1. The question about the sample size remains. The authors replied that they did not calculate the sample size as they decided to include all dentists affiliated with the mentioned association. However, the stated response and participation rates were very low, around and less than 40%. This brings up the issue of whether this study was powered enough to detect meaningful and significant differences. If you state that "Considering sample size calculation, we did not calculate the sample size in this cohort study and decided to include all the dentists affiliated with the Aichi Dental Association." and you have a participation rate of 41.6% then that is not a satisfactory sample size for this type of study.

Response: We regret that we were not able to adequately consider this. In response to your suggestion, we calculated the sample size using the EZR statistical software. As a result, the required sample size was satisfied, although the participation rate was approximately 40%. In addition, we have added the sample size calculation subsection (p. 2, lines 80−84) and reference #17 (p. 10, lines 354−355) in the revised manuscript.

  1. Regarding the question of missing data, and in line with your response - you defined a questionnaire as complete if it contained complete information on conventional smoking status, but what about all of the other questions? Explain.

Response: Thank you for pointing this out. The missing data of other questions are as follows: age (n = 3), dental education level (n = 29), state of employment (n = 3), and HTP use status (n = 293). We have also added the number of these non-responders to the table legend (p. 4,5, lines 142−145, lines 150−152, Tables 2 and 4).

  1. Results: Almost no changes were made to the text, and none in line with the received comments regarding the description of results given on tables, nor was a rationale provided.

Response: The details of the Tables are explained as follows:

Considering Tables 2 and 3, exclusive smokers/users exist among current smokers/users; the data have not changed. However, since the sentence was misleading, the tables have been revised to make it easier to understand, listing details regarding the current smokers/users, exclusive smokers/users, and dual users (p. 5, lines 146−152, Tables 3 and 4).

To avoid misinterpretation of the samples, the table was modified by adding the number of no response to the table legend.

Additionally, we have revised the conventional cigarette smoking status (p.6, lines 158−159) and HTP use status (p.6, lines 165−166) subsections to focus on the age and sex distribution and showed that there were statistically significant differences in the revised manuscript.

  1. My comment on the original version of the manuscript was: "Line 177: Explain the notable differences in cigarette smoking prevalence in this study in comparison to others.". The Authors' reply is that they added the word "far" to another sentence. This does not represent a response to this comment. No explanation was provided. Simply listing results of this and other studies does not represent a discussion nor provides any meaningful information. Providing possible explanations for any noted differences is the main part of the section discussion.

Response: We regret that we were unable to adequately correct the problem despite your suggestion. To make this point clearer, we have added the following sentence to the Discussion section of the revised manuscript (p. 7, lines 197−199): In addition to the fact that more than 90% of the dentists surveyed were male, this group had a significantly higher percentage of male smokers, which may have contributed to the higher smoking rate compared to other medical professionals.

  1. My comment on the original version of the manuscript was: "Line 180-181: Explain `trends` and where in this manuscript it was described.". The authors responded by adding two references. This is misleading - the sentence states that the present study analyzed trends - explain where.

Response: We regret that we were unable to adequately correct the problem despite your suggestion. We have avoided the term "trend" to prevent misinterpretation and instead revised it as " the reality of complex forms of smoking, including HTP " (p. 7, lines 204−205). The references were also cited to support the fact that awareness of HTPs in Japan has generally increased in recent years.

  1. My comment on the original version of the manuscript was: "Line 203: Special attention in the section Discussion should be paid to age and sex distribution, both in conventional and HTP use, where statistically significant differences were presented in this manuscript. Give a comparison with results of other studies and an explanation for the found differences." was not answered appropriately, even though authors stated they did. No explanations were provided, and comparisons are made with just one study even though "studies" are mentioned by the authors.

Response: We apologize for not responding appropriately despite your suggestion. To make this point clearer, we have added the following sentence to the Discussion section of the revised manuscript (p. 7, lines 208−212): For this reason, in addition to the tendency of those with conventional cigarette smoking experience to initiate HTP use, we considered that the results were influenced by the fact that both conventional cigarette and HTP were significantly smoked by males aged 50−59 years, a group that smoked significantly more than the other groups.

  1. In a response to my comment, authors stated that they: "Moreover, we have added a new limitation, which was the absence of potential confounders such as socioeconomic status, education level, etc.". However, no changes were made to the section Limitations - this is misleading.

Response: We apologize for this misleading expression. To clarify, we have added the following sentence to the Study limitation (p. 9, lines 276-279): Third, the respondents in this study were only dentists with clinical experience, most of whom were men, which was due the absence of potential confounders, such as socioeconomic status, education level, etc. This may have influenced the results. We believe that this new information adequately addresses the reviewer’s comment.

  1. The Authors' response to the last comment regarding the Conclusion is extremely misleading

Response: We regret that we were unable to adequately correct the problem despite your suggestion. We have reviewed your comments again and deleted the first two sentences because they are misleading. (p. 9, line 289).

  1. the authors completely ignored the provided remarks, interpreted them wrongly in a way that enabled them to make no changes other than reorder the words and add a word to make it seem as if they have allegedly answered the comment. The issue still remains - instead of underlining their own results, the authors make unjustified comparisons and conclusions that are no where related to their results.

Response: We sincerely apologize that, despite your suggestion, we overlooked the problems and were unable to adequately respond to and correct them. We hope that the resolution offered will be acceptable to you. We value your feedback and are grateful for the opportunity to rectify the issues noted.
